# Prostanoid Metabolites as Biomarkers in Human Disease

**DOI:** 10.3390/metabo12080721

**Published:** 2022-08-04

**Authors:** Helena Idborg, Sven-Christian Pawelzik

**Affiliations:** 1Division of Rheumatology, Department of Medicine, Solna, Karolinska Institute, Karolinska University Hospital, SE-171 76 Stockholm, Sweden; 2Cardiovascular Medicine Unit, Department of Medicine, Solna, Karolinska Institute, SE-171 76 Stockholm, Sweden; 3Division of Vascular and Coronary Disease, Theme Heart and Vessels, Karolinska University Hospital, SE-171 76 Stockholm, Sweden

**Keywords:** prostanoid, prostaglandin, prostacyclin, thromboxane, eicosanoid, biomarker, metabolism, creatinine, LC–MS/MS

## Abstract

Prostaglandins (PGD_2_, PGE_2_, PGF_2__α_), prostacyclin (PGI_2_), and thromboxane A_2_ (TXA_2_) together form the prostanoid family of lipid mediators. As autacoids, these five primary prostanoids propagate intercellular signals and are involved in many physiological processes. Furthermore, alterations in their biosynthesis accompany a wide range of pathological conditions, which leads to substantially increased local levels during disease. Primary prostanoids are chemically instable and rapidly metabolized. Their metabolites are more stable, integrate the local production on a systemic level, and their analysis in various biological matrices yields valuable information under different pathological settings. Therefore, prostanoid metabolites may be used as diagnostic, predictive, or prognostic biomarkers in human disease. Although their potential as biomarkers is great and extensive research has identified major prostanoid metabolites that serve as target analytes in different biofluids, the number of studies that correlate prostanoid metabolite levels to disease outcome is still limited. We review the metabolism of primary prostanoids in humans, summarize the levels of prostanoid metabolites in healthy subjects, and highlight existing biomarker studies. Since analysis of prostanoid metabolites is challenging because of ongoing metabolism and limited half-lives, an emphasis of this review lies on the reliable measurement and interpretation of obtained levels.

## 1. Introduction

The prostanoid family of bioactive lipid mediators comprises five members, prostaglandin (PG)E_2_, PGD_2_, PGF_2__α_, prostacyclin (PGI_2_), and thromboxane A_2_ (TXA_2_). These five prostanoids mediate biological function and are therefore denoted as primary prostanoids. They are effective as bioactive lipid mediators and involved in many critical physiological and pathophysiological processes, among others, blood pressure homeostasis [1,2,3], sleep regulation [4,5], labor [6], induction of fever [7,8,9,10], and pain [11,12] as well as inflammatory reactions [13,14,15,16] and malignancies [10,17,18]. Although their functions have been studied for decades, a complete understanding of their importance in health and disease is still elusive due to the complexity of their signaling. Their locally and temporally restricted biosynthesis, which is limited by the expression of their synthesizing enzymes to usually one or two prostanoids per cell type, causes complex profiles that are specific for each tissue. Furthermore, target cells can express multiple types of prostanoid receptors with sometimes opposing downstream functions, which contributes to sophisticated signaling. The importance of primary prostanoids as mediators was previously empathized and recently reviewed with a focus on enzyme [19,20] and receptor [21] expression, respectively. Another layer that contributes to the complexity of prostanoid signaling is the fact that primary prostanoids are short lived under physiological conditions. Rapid metabolic conversion of prostanoids leads to their functional inactivation and thus restricts their biological action to auto- or paracrine effects in the immediate vicinity of their origin.

Metabolites of primary prostanoid are usually biologically inactive, but may integrate changes in the local prostanoid levels and may therefore be useful biomarkers to indicate human disease. In this review, we survey the metabolism of each of the primary prostanoids, identify the major metabolites and their concentrations in different biofluids of healthy human subjects, and evaluate their stability in vivo and in vitro with respect to their suitability as biomarkers. Although our emphasis lies on frequently sampled biofluids such as plasma and urine, we also review other specimens in which prostanoid metabolites were measured. We give examples of prostanoid metabolites that were studied in the context of human disease and suggest interpretations of what their levels in different body compartments may reflect. We furthermore comment on analytical approaches for the detection of prostanoid metabolites and give practical guidance that will help to expand the use of prostanoid metabolites as biomarkers for human disease.

## 2. Metabolism of Prostanoids

Prostanoids are derived from the ω-6 polyunsaturated fatty acid arachidonic acid (AA; 20:4^Δ5Z,8Z,11Z,14Z^). Cyclooxygenase (COX), the key enzyme in the biogenesis of prostanoids, converts free AA in a two-step enzymatic reaction to form the labile intermediates PGG_2_ and PGH_2_ (Figure 1). The intermediate PGH_2_ is the common substrate to various terminal synthases, which give rise to the five primary prostanoids. The enzymatic mechanism of this catalysis was studied in detail [22,23,24,25], and the anabolism of prostanoids was previously reviewed [19,20,26]. As a consequence of their shared molecular origin, all primary prostanoids are characterized by a fatty acid backbone of 20 carbon atoms and thus belong to the class of eicosanoid signaling molecules in which they constitute their own subfamily. All primary prostanoids, with the exception of TXA_2_, also maintain a cyclopentane ring, which is formed by COX and is a structural characteristic of prostaglandins.

Figure 1 provides, in a nutshell, a schematic overview of the catabolic pathways involved in the metabolism of each primary prostanoid and identifies the major metabolites in plasma and urine, respectively. Besides these major metabolites, several other metabolites can be detected at different levels in the various specimens. Their concentrations and basic characteristics are summarized in Table 1, and Appendix A lists their molecular identity, including molecular structure, as well as possible synonym names that were used throughout the literature.

The metabolic fate of primary prostanoids in humans was investigated by infusion studies of radioactively labeled molecules [28,37,50,51]. PGE_2_, PGD_2_, and PGF_2α_ have half-lives in the circulation that are reported to be in the range of minutes [24,28,52] (Table 1). The ubiquitously expressed enzyme 15-hydroxyprostaglandin dehydrogenase (15-PGDH) [27] oxidizes these three prostanoids at the hydroxyl group at C-15 and thereby renders them biologically inactive. PGI_2_ and TXA_2_ follow a somewhat different path with even shorter half-lives than that of the other prostanoids, which were reported to be in the range of seconds [39] (Table 1). Due to their chemical instability, these two prostanoids decompose non-enzymatically to form the biologically inactive metabolites 6-keto PGF_1α_ and TXB_2_, respectively. PGE_2_, PGD_2_, and PGF_2__α_ are, therefore, under certain conditions, amenable to analytical methods, while 6-keto PGF_1__α_ and TXB_2_ are almost always measured as direct substitutes for PGI_2_ and TXA_2_, respectively. For analytical purposes and throughout this review, primary prostanoids are thus referred to as PGE_2_, PGD_2_, PGF_2__α_, 6-keto PGF_1__α_ and TXB_2_.

### 2.1. Metabolites of PGE_2_

PGE_2_ has a biological half-life of about 1.5 min [28,52] and is metabolically inactivated by 15-PGDH to form 15-keto PGE_2_, which is subsequently reduced at the double bond between C-13 and C-14 by 15-oxo-prostaglandin Δ13-reductase [53]. Its primary plasma metabolite is 13,14-dihydro-15-keto PGE_2_ (Figure 2), which accumulates to detectable levels in the range of about 20 pg/mL 32. Although 13,14-dihydro-15-keto PGE_2_ has a longer plasma half-life than PGE_2_ of about 9 min, it undergoes further metabolism and is therefore rather unreliable as a measure to estimate the production of PGE_2_ in humans [29,32]. One of the catabolic steps in its further metabolism is the non-enzymatic dehydration of 13,14-dihydro-15-keto PGE_2_ to form 13,14-dihydro-15-keto PGA_2_ [30,54]. This reaction is facilitated by albumin, which is abundant in plasma and thus yet another factor for the rapid metabolism of prostanoids. Both 13,14-dihydro-15-keto PGE_2_ and 13,14-dihydro-15-keto PGA_2_ can be converted to the stable, base-catalyzed breakdown product, bicyclo PGE_2_, to integrate the intermediates that are subjected to ongoing metabolism 29. Bicyclo PGE_2_ should be quantified to reflect plasma levels of PGE_2_ [54,55].

Further metabolic pathways of PGE_2_, but also PGE_1_, involve the removal of four carbons at the α-terminus via β-oxidation and additional oxidation of the terminal ω-carbon to yield tetranor-PGEM, which is excreted by the kidneys as the major urinary metabolite of PGE_2_. Tetranor-PGEM is commonly used as a surrogate marker to estimate the systemic biosynthesis of PGE_2_ [56,57]. The levels of tetranor-PGEM in healthy humans range between 2.6 and 7.4 ng/mg creatinine but can be doubled under pathological conditions [31] (Table 1).

### 2.2. Metabolites of PGD_2_

PGD_2_ has an approximate plasma half-life of 0.9 min [33] (Table 1). It gives rise to different metabolites that either retain the D-series cyclopentane ring with a hydroxy group at position C-9 and a keto group at position C-11, adopt the F-series ring with a hydroxy group at both positions, or adopt the cyclopentanone ring of the J-series (Figure 3). In vivo studies on the metabolic fate of radiolabeled, exogenous PGD_2_ revealed 25 different urinary metabolites; 23 of which belonged to the F-series [50]. The major human metabolite, however, belonged to the D-series [50]. This major metabolite, tetranor-PGDM, is metabolized analog to PGE_2_ via the 15-PGDH pathway to initially form 13,14-dihydro-15-keto PGD_2_ in plasma [58] before undergoing β- and ω-oxidation and being excreted in the urine [59]. Tetranor-PGDM reflects the systemic biosynthesis of PGD_2_ in vivo. Its concentration in the urine of healthy human subjects is low; between 0.3 and 2.5 ng/mg of creatinine [31] (Table 1).

Reduction of the keto group of the cyclopentane ring of PGD_2_ yields 11β-PGF_2α_ and is catalyzed by the NADPH-dependent enzyme 11-ketoreductase [60], which is expressed in the lungs [61] and the liver [60].

In an aqueous solution or in plasma, PGD_2_ degrades non-enzymatically to yield Δ^12^-PGD_2_ and PGJ_2_, respectively [33]. This transformation is reinforced by plasma albumin. Further non-enzymatic degradation gives rise to their respective downstream metabolites, 15-deoxy-Δ^12,14^-PGD_2_, Δ^12^-PGJ_2_, and 15-deoxy-Δ^12,14^-PGD_2_ (Figure 3). The formation of Δ^12^-PGJ_2_ from PGJ_2_ strictly depends on albumin [33].

Unlike the metabolites of all other primary prostanoids, many of the PGD_2_ metabolites retain biological activity, although not via the PGD_2_ receptors DP1 and DP2 [62]. 11β-PGF_2α_ was shown to contract bronchial smooth muscle cells [63] as well as coronary arteries [64] and induce phosphorylation of the serine/threonine-specific protein kinase ERK [65]. 15-deoxy-Δ^12,14^-PGD_2_ was shown to activate the nuclear receptor PPARγ in macrophages and has anti-inflammatory properties [66]. Δ^12^-PGJ_2_ was assigned anti-tumor and anti-viral properties [67]. 15-deoxy-Δ^12,14^-PGJ_2_ also ligates PPARγ and promotes the differentiation of fibroblasts to adipocytes [68,69]. The mechanism of action of the cyclopentenone prostanoids of the A and J series was previously reviewed [70].

Although the demonstrated biological activity of PGD_2_ metabolites required much higher concentrations than those measured in physiological samples, the local tissue concentrations may differ markedly from the circulating levels, among others due to adhesion to albumin or conjugation with glutathione. For instance, while plasma and urinary levels of 15-deoxy-Δ^12,14^-PGJ_2_ in healthy human individuals were reported in the range of 2 to 350 pg/mL [34] and 6.3 ± 2.7 pg/mg of creatinine [35], its tissue levels of 15 µM (4.7 µg/mL) are magnitudes higher and similar to those of arachidonic acid in tissues [53].

### 2.3. Metabolites of PGF_2__α_

The same pathway that metabolizes PGE_2_ and PGD_2_ converts also PGF_2__α_ to form, through modification by 15-PGDH and 15-oxo-prostaglandin Δ13-reductase, its initial, biologically inactive plasma metabolite, 13,14-dihydro-15 keto PGF_2__α_ (Figure 4) [71,72]. 13,14-dihydro-15 keto PGF_2__α_ accumulates in human plasma to concentrations of 0.08–20 pmol/mL [36] but prevails only briefly. Further metabolism involves β- and ω-oxidation and leads to tetranor-PGFM [73], which is detectable in urine, but also in plasma [74]. In fact, about 20 min after injection of radiolabeled PGF_2__α_, tetranor-PGFM was the dominating plasma metabolite and stayed longer in the circulation than its precursor 13,14-dihydro-15 keto PGF_2__α_ [37]. It was therefore suggested to preferably measure tetranor-PGFM as a more reliable plasma biomarker. PGF_2__α_ is vastly produced by females during labor. Consequently, the plasma levels of 13,14-dihydro-15 keto PGF_2__α_ rise from low basal levels of about 40–60 pg/mL during late pregnancy to peak concentrations of 1200–4100 pg/mL and decrease rapidly to almost baseline levels 2 h after parturition. Interestingly, plasma levels of tetranor-PGFM follow a slightly different pattern and increase from basal levels of about 60–100 pg/mL during late pregnancy to reach a peak concentrations between 1000 and 2000 pg/mL; however, unlike 13,14-dihydro-15 keto PGF_2__α_, tetranor-PGFM peaks later, about 2 h after parturition, and its levels stay at these high concentrations for several hours and remain elevated (about 100–300 pg/mL) 24 h after parturition [37].

Tetranor-PGFM is excreted by the kidneys and constitutes the major urinary metabolite of PGF_2__α_. The mean urinary concentration of tetranor-PGFM in healthy subjects is 1.2 ± 7.1 µg/24 h in urine for women and 1.6 ± 6.0 µg/24 h in urine for men [38].

### 2.4. Metabolites of TXA_2_

TXA_2_ is an unusual bicyclic prostanoid, displaying an oxane ring instead of the cyclopentane ring that is characteristic of the other four prostanoids, coupled to a cyclic ether. TXA_2_ is released in substantial quantities from aggregating platelets and is highly unstable under physiological conditions. The cyclic ether function on its oxane ring is non-enzymatically hydrated in aqueous solution within about 30 s [39,75] to form a hemiacetal and thus the biologically inactive TXB_2_ (Figure 5). The levels of TXB_2_ are very low in plasma from uninduced blood, around 1–2 pg/mL (3–6 fmol/mL), in line with TXA_2’s_ pro-aggregatory, vaso-, and bronchoconstrictive function. However, these low levels can increase tremendously during hemostasis to 300–400 ng/mL (0.8–1.0 nmol/mL) in serum from fully coagulated blood [40]. While TXB_2_ is relatively stable with a half-life of about 7 min [41] and readily detectable in plasma and urine, it is not necessarily well suited as a biomarker to assess in vivo TXA_2_ formation due to its further metabolism.

There are a total of twenty metabolites of TXB_2_ that can be detected in urine [51]; however, two pathways were identified to account for the majority of metabolites. The first pathway is the direct oxidation of TXB_2_ by 11-hydroxy Thromboxane Dehydrogenase and leads to 11-dehydro TXB_2_ [76,77]. The second major pathway is the β-oxidation of the carboxy terminus of TXB_2_, leading to 2,3-dinor-TXB_2_ [78].

Both metabolites are detectable in plasma and urine; however, 11-dehydro TXB_2_ was identified as the main plasma metabolite while 2,3-dinor TXB_2_ was the main urinary metabolite [79]. In plasma, 11-dehydro-TXB_2_ has a half-life of 45–60 min and accumulates to levels of 0.9–4.3 pg/mL. Its levels in urine are about 70 ng/mmol of creatinine [80]. The median level of excreted 2,3-dinor TXB_2_ in the urine ranges from 138 pg/mg of creatine (10.3 ng/h) in healthy male volunteers [78] to about 175 pg/mg creatinine in healthy subjects of both sexes [49]. 2,3-dinor TXB_2_ displays neither a significant sex difference nor any significant diurnal variation [49].

### 2.5. Metabolites of PGI_2_

PGI_2_ is also a chemically unstable bicyclic molecule and has a similar short half-life in vivo as compared to TXA_2_. The vinyl ether moiety between C-6 and C-9 of PGI_2_ is hydrolyzed to form 6-keto PGF_1α_ within about 30 s in vivo. Ex vivo, PGI_2_ is substantially more stable with a half-life of about 6–10 min in whole blood or plasma [45], and in pharmacological preparations at basic pH, PGI_2_ is stable for at least 48 h [81]. 6-keto PGF_1α_ is biologically inactive and with a plasma half-life of about 30 min [81] relatively stable and can be used as a biomarker for the PGI_2_ biosynthesis in vivo [82]. Plasma concentrations of 6-keto PGF_1α_ in healthy individuals were determined to be 1.9 ± 0.8 pg/mL [48].

6-keto PGF_1α_ can not only be detected in human plasma, but also in human urine; however, the levels of urinary 6-keto PGF_1α_, which were determined to be 92 ± 51 pg/mL, or, as reported normalized to creatinine, as 168 ± 91 pg/mg creatinine [48], account for only about 6% of all urinary metabolites of PGI_2_ in healthy volunteers [82] and were suggested to reflect renal production of PGI_2_ rather than its systemic biosynthesis [83]. Further metabolism of 6-keto PGF_1α_ yields at least 16 compounds, which are primarily excreted in the urine [82]. The major urinary metabolite in humans is 2,3-dinor-6-keto PGF_1α_ (Figure 6) [84], which was found to be 3-fold more abundant than 6-keto PGF_1α_ [48]. The average excreted level of 2,3-dinor-6-keto PGF_1α_ in healthy individuals is around 100 pg/mg creatinine. There is neither a significant sex difference in the excreted levels of 2,3-dinor-6-keto PGF_1α_ nor any significant diurnal variation [49]. Urinary 2,3-dinor-6-keto PGF_1α_ was thought of as a marker for systemic production of PGI_2_ [85]; however, this concept has been recently challenged [86].

## 3. Analysis of Prostanoid Metabolites

It is critical to select appropriate biological samples and sufficiently stable target analytes when designing a study or clinical trial to obtain meaningful and robust results. A good understanding of the pathway and detailed knowledge about the continuous metabolism of primary prostanoids, especially regarding the expression and location of metabolizing enzymes as well as excretion of the respective metabolites, is crucial to making the right choices. Furthermore, proper short- and long-term storage of the samples and handling during the analytical process is critical to ensure that all samples of a given cohort that are collected over a longer period are treated equally and analyte losses are avoided.

### 3.1. Sample Selection and Choice of Analyte(s)

In many non-physiological matrices devoid of 15-PGDH and 15-oxo-prostaglandin Δ13-reductase, e.g., in the supernatant of cultured cells, metabolism of primary prostanoids can be excluded. Under these conditions, PGE_2_, PGD_2_, and PGF_2α_ are relatively stable and may be readily measured to estimate the biosynthetic capacity of the respective cells in vitro [16,87,88]. 6-keto PGF_1α_ and TXB_2_, the immediate non-enzymatical breakdown products of PGI_2_ and TXA_2_, respectively, serve under these circumstances as surrogate markers [87]. These prostanoids were also measured in samples of exhaled breath condensate [89] and in tissue samples [90] instead of their downstream metabolites.

Plasma and serum on the other hand are generally problematic biological samples for the analysis of prostanoids. Although some metabolites can be detected in these matrices [29,32], their levels will change over time due to ongoing metabolism, resulting in a variable temporal flux that will influence the study results. Therefore, although blood samples are frequently analyzed for other biomarkers, one should avoid measuring prostanoids in plasma or serum, respectively if other options are available. One noticeable exception from this general rule is the ex vivo analysis of serum TXB_2_, which has proven useful to assess COX-1 dependent formation of TXA_2_ in fully coagulated blood and its inhibition by Aspirin, respectively [40].

A more reliable source to measure and interpret in vivo formation of prostanoids is urine. Unlike the primary prostanoids with auto- or paracrine biological activity but a very limited operating range, the urinary prostanoid metabolites have undergone several steps of metabolism during which they have lost their biological activity, but as final metabolic products, they are relatively stable and integrate systemic biosynthesis. Therefore, it is very informative to quantify their levels in disease as compared to the healthy state or changes in their levels that were caused by pharmaceutical intervention.

### 3.2. Sample Collection, Handling, and Storage

Accurate quantification of prostanoids relies on many factors. Correct sampling, handling, and short- as well as long-term storage, will contribute to the accuracy of the analysis results, reflecting in part the unstable nature of prostanoids.

Avoiding artifacts during sample collection is of particular importance for the analysis of 6-keto PGF_1__α_ and TXB_2_, respectively, in blood samples. Venipuncture may activate endothelial cells to produce PGI_2_, leading to artifactually increased levels of 6-keto PGF_1α_ in plasma samples, which reflects incorrect sample handling rather than real biosynthesis [91]. Similarly, platelets are easily activated ex vivo during the blood draw, leading to erroneously increased levels of TXB_2_ in the samples [42]. For more reliable measurements of the systemic TXA_2_ formation, analysis of downstream metabolites was suggested [41,92]. In this context, and as discussed below in Section 4, the consumption of Aspirin and other Non-Steroidal Anti-Inflammatory Drugs (NSAIDs) should be closely monitored in the subjects because these over-the-counter drugs hamper prostanoid biosynthesis, and are widely used but oftentimes not considered genuine drugs by the study participants and therefore not always correctly declared. Due to its irreversible inhibition and preferred binding to platelet COX-1, even low doses of Aspirin have a profound cumulative and long-lasting effect on the formation of TXA_2_ [93].

While sampling artifacts can lead to increased levels of TXB_2_, inappropriate storage after sample collection may result in substantial losses of the analytes of interest due to their chemical instability and subsequent decomposition. For example, storage of frozen urine samples at −20 °C resulted already after only one week in significantly decreased levels of tetranor-PGEM and losses of up to 80% of the initial amounts after 1.5 to 2 years [8]. On the contrary, the tetranor-PGEM levels in urine samples that were stored at −80 °C instead for the same time were indistinguishable from samples that were quantified directly after sampling [8]. This observation is particularly important for large cohort studies with samples collected at different time points. To avoid inconsistencies and false results, a few principles should be considered: the time from sampling to freezing should be minimized; samples should be aliquoted to avoid repetitive freeze–thaw cycles; ideally, the volume of the aliquots corresponds to the volume that will be later analyzed, and authentic internal standards are added directly at the time of freezing; and the samples should be stored at low temperatures. If possible, a pilot study should be run as part of a proper method validation before the study or clinical trial begins to characterize the stability of the analyte(s) of interest.

### 3.3. Methodology for the Analysis of Prostanoid Lipid Mediators

Enzyme immunoassays (EIA) were extensively used to analyze primary prostanoids as well as their downstream metabolites. Immunoassays combine the advantage of high sensitivity and thus the ability to detect minute analyte levels with a fairly easy implementation that does not require advanced detection technology or specialized analytical skills. They are commercially available for many prostanoid metabolites, but not for all, and they are usually very reliable and reproducible. Furthermore, immunoassays have the advantage that they are amenable in the analysis of large sample numbers. Therefore, this method is still frequently employed. However, immunoassays do not provide a direct measurement of the analyte but rather an indirect measurement of a tracer molecule. Moreover, they are not amenable for the analysis of several analytes at the same time, and the quality of each immunoassay is highly dependent on the specificity of the implemented antibody. This may result in a loss of accuracy due to cross-reactivity with other compounds than the target analyte, which is a particular concern in the analysis of structurally similar prostanoids.

Nowadays, liquid chromatography coupled with tandem mass spectrometry (LC–MS/MS) is regarded as the gold standard method for the detection of prostanoids and downstream metabolites. Mass spectrometry was the key methodology in the discovery of prostanoids, their structural elucidation [94], and early quantitation studies [95,96]. Mass spectrometers are broadly applicable to many analytes in biological matrices, they are very sensitive and highly accurate, especially when several characteristic ion transitions per analyte are scrutinized in a typical multiple reaction monitoring (MRM) method. Furthermore, it is possible to simultaneously detect and quantify several target analytes in the same sample, and with advanced LC–MS/MS methods with optimized chromatography, it is even possible to screen larger cohorts [97]. Hence, LC–MS/MS may be approaching the medium to high throughput capacity of immunoassays while gaining accuracy, specificity, and the number of analytes.

In LC–MS/MS, the analyte molecules are separated in an initial chromatographic step based on their interaction with a stationary phase according to their chemical properties. This separation step is crucial, especially when constitutional isomeric prostanoids with identical mass-to-charge ratios such as, e.g., PGE_2_ and PGD_2_ or tetranor-PGEM and tetranor-PGDM are among the target analytes [98,99]. Analyte identification is based on a characteristic combination of the retention time during the chromatography step and the specific mass transition(s) in the MRM tandem mass spectrometry step, which takes into account the molecular masses both of the precursor ion and one or several fragment ions. This ensures the high specificity and accuracy of the method. In real life, however, only one transition pair might be monitored or only the dominant fragment ion may be detectable at the low concentrations in the samples. Longer LC gradients generally decrease the risk of wrongly assigned analyte peaks but decrease sample throughput at the same time. In addition, chiral chromatography might be needed to separate stereoisomers. This includes the separation of enzymatically and non-enzymatically formed species. Biological samples may undergo oxidation during prolonged storage, which may create prostanoid-like, non-enzymatically formed isoprostanes [100]. Difficulties that may be encountered in LC–MS/MS methods, including unique MRM transitions, in-source fragmentation, and matrix effects were previously discussed [98].

Some prostanoids, such as TXB_2_ or 2,3-dinor-6-keto PGF_1__α_, produce several tautomeric forms, which appear as a conglomerate of individual signals and cause a broad peak shape during chromatographic separation. While quantification of these tautomers is usually unproblematic at sufficiently high concentrations, splitting a small signal at lower levels into several tautomeric peaks may result in a broad but shallow signal that falls below the required signal-to-noise ratio (S/N) of 10 for reliable quantification. For these compounds, derivatization may help to avoid tautomerization and enhance sensitivity [101].

The levels at which prostanoids are present in human biological samples are in the pmol to fmol range, making the sensitivity of the LC–MS/MS instrument an important feature to obtain a low limit of quantification. Sensitivity may be further enhanced by an initial sample preparation that purifies and concentrates the analytes of interest. Proper sample preparation can be a challenging task for a method that aims to quantify several metabolites in a single run and reach a decent overall sensitivity for all metabolites. Depending on chemical properties and interaction with different solvents and sorbents used in liquid–liquid extraction (LLE) or solid phase extraction (SPE), respectively, as well as depending on method optimization, losses may be more than 50% for individual analytes. Therefore, deuterated internal standards for every monitored analyte should be included in all methods that involve extraction to account for losses during sample preparation and compensate for matrix effects such as ion suppression [102]. The internal standard should be added as early as possible in the sample preparation procedure, and incubation time in the biological matrix should be long enough to be able to compensate for possible protein binding.

## 4. Frequently Encountered Obstacles When Analyzing Prostanoid Metabolites

### 4.1. Accuracy of Analytical Results Obtained by Immunoassay versus LC–MS/MS

Enzyme immunoassays and mass spectrometry-based assays are the two prime methodologies that are employed for the measurement of primary prostanoids and their metabolites as well as to quantify other lipid mediators. When comparing the literature values, it is eye-catching that results obtained by immunoassays are usually several fold higher than those obtained by LC–MS/MS [31]. A study that measured urinary 2,3-dinor-5,6-dihydro-15-F_2t_-isoprostane and directly compared the obtained results from two different methods found on average about 30-fold greater values when using an immunoassay as compared to gas chromatography (GC)–MS/MS analysis and an overall poor correlation (Pearson’s r = 0.51) between the two methods [103]. The tendency to overestimate analyte levels with immunoassays is most likely characteristic of the inherent weaknesses of this method, i.e., lack of analyte clean-up due to missing sample separation combined with a varying degree of antibody specificity that may lead to cross-reactivity with related compounds. Therefore, the measured levels may rather represent the sum of several metabolites than only a single target analyte and thus appear higher than their real levels because they are quantified based on the comparison to clean, synthetic, external standards. This may be a subsidiary analytical problem as long as relative changes in samples within one study are regarded that are all handled and measured by the same assay; however, it prevents the comparison of absolute values between studies. On the other hand, immunoassays of good quality, employing highly specific antibodies may well be able to take on results obtained by mass spectrometry [16]. Furthermore, biological samples may undergo oxidation during storage and sample handling, leading to isoprostanes, which are non-enzymatically formed, prostanoid-like molecules that can interfere with enzyme immunoassays and LC–MS/MS analysis, and result in measuring artifacts [100].

### 4.2. Undisclosed Use of NSAIDs May Compromise Study Results

When analyzing prostanoid metabolites as biomarkers in disease the biosynthetic pathway leading to their generation should be considered. COX is the molecular target of the widely used class of NSAIDs [13,104], which exert their analgesic, antipyretic, and anti-inflammatory effects through suppression of prostanoid biosynthesis, in line with the prominent role of prostanoids in the signaling of pain, inflammation, and the induction of fever. In 2010, about 12.8% of adults in the US consumed NSAIDs on a regular basis. An additional 19.0% of US adults chronically consumed low-dose aspirin for secondary prevention of cardiovascular disease [105]. Therefore, the levels of prostanoid metabolites measured in a given cohort may be underestimated due to inhibition of their biosynthesis in a substantial part of the general population, and the use of NSAIDs should be carefully controlled in any study that aims to quantify prostanoid metabolites as biomarkers for disease. However, non-reported NSAID usage may be unavoidable.

### 4.3. Normalization of Urinary Metabolites

Most biological fluids that may serve as samples for the analysis of prostanoid metabolites, including plasma, serum, and cerebrospinal fluid (CSF), are tightly controlled in their physiological composition. Therefore, mere concentrations of analyte per volume may suffice to compare different individuals, cohorts, or even studies from different labs. Urine, on the other hand, the biological fluid that contains the most stable prostanoid metabolites, which hence constitutes the best-suited biomarkers for disease, may differ substantially from sample to sample. This depends largely on the intake of water, which affects the total volume of excreted urine and thus the concentration of solutes. As a result, the investigation of specific target analytes requires normalization to compensate for these effects.

Among the most widely accepted normalization approaches are the use of urinary creatinine concentrations and urinary osmolality. These two methods yield comparable results; however, both methods have also certain drawbacks [106], and other normalization approaches might be better suited for specific study populations [8]. In the best analytical scenario, several normalization approaches can be used to yield similar results and thus validate the results [107].

Measurement of urinary analytes as a ratio to the creatinine concentration is considered the gold standard in normalization. Ideally, creatinine is released at a constant rate from muscle and protein metabolism and cleared from the blood at a constant rate in healthy patients with regular kidney function. Hence, the urinary creatinine concentration depends solely on the total volume of excreted urine and is therefore suitable for volume corrections of any other solute in the same sample. This implies, however, a healthy renal function and a constant glomerular filtration of creatinine. Moreover, since creatine release into the blood depends, among other factors, on the muscle mass (which generally differs between sexes), exercise status, and age of the subjects, the groups that are to be compared in a given study need to be carefully matched with regard to these parameters, and kidney function, as well as possible differences in creatinine excretion, need to be considered during the conception of a study.

Osmolality measures the urine concentration by considering all solutes that contribute to the osmotic pressure of a given urine sample and thus lower its freezing point compared to water. The above-mentioned factors that may compromise urinary creatinine concentrations do not affect osmolality [107]. However, a freezing point depression osmometer is necessary to determine osmolality, which is not available in all laboratories. Furthermore, insoluble particles or inhomogeneous samples may hamper accurate measurements of osmolality.

Alternative normalization strategies include a consistent and uniform dilution of all urine samples based on their optical absorption at 300 nm [101] and the use of the total useful MS signal [107], which requires elaborate equipment but is easily available when the analysis is performed using LC–MS/MS. In infants, whose kidney function is not yet fully developed and hence creatinine concentrations are not meaningful, the body surface area (BSA) was used before to normalize for metabolic status and excretion attainment [8].

### 4.4. Collection of 24 h Urine versus Spot Urine Samples

Initially, urinary prostanoid metabolites were studied in 24 h urine [108]. However, this method is intricate, relies on good subject compliance, and limited stability of some metabolites [8] at storage conditions available during the sampling period may impair the results. To circumvent these difficulties, normalization methods allow the use of spot urine samples instead because analytes that are normalized to, e.g., creatinine accurately reflects the 24 h excretion of the respective analyte. For instance, the creatinine-normalized levels of 2,3-dinor-6-keto PGF_1α_ and 2,3-dinor-TXA_2_ in spot urine samples from 3 h long time intervals during a 24 h sampling period correlated highly with the total amounts determined for the respective interval. Simple concentrations in these spot urine samples, on the other hand, were poor indices of the 24 h excretion of both analytes [49]. General directions on how to sample, handle, store, and prepare urine samples prior to analysis were thoroughly reviewed [102].

### 4.5. Renal versus Extrarenal Origin of Urinary Prostanoid Metabolites and Data Interpretation

For reasons discussed in this article, i.e., rapid and continuous metabolism of prostanoids in vivo, the measurement of excreted stable end products of prostanoid metabolism has proven to be a useful approach. Thus, urine has lent itself as a valuable and informative source of biological samples. However, to judge what the measured levels really reflect in a given biological context, a careful interpretation of the results is demanded. Biosynthesis, metabolism, and excretion happen continuously and in parallel; therefore, many of the prostanoid metabolites are detectable both in plasma and in urine (Table 1). Here, we have reviewed the metabolites that are commonly regarded and measured as the major ones in the respective biofluids; however, most of the minor metabolites can also be detected in these samples. Analysis of several metabolites at the same time, which is easily achieved when using LC–MS/MS methodology, may serve as an internal verification of the results or even yield additional information and address thought-provoking aspects of the research question.

When analyzing and interpreting data from urine samples, one needs to carefully consider that this is an indirect way to estimate the systemic generation of prostanoids in vivo. Biosynthesis of prostanoids is locally restricted and occurs in a cell- and tissue-specific way. Prostanoid metabolites detected in the urine integrate biosynthesis of prostanoids in the entire organism but do not indicate their origin. Kidney function is particularly important when analyzing prostanoid metabolites in the urine; not only for its excretion activity, which affects both the analyte and the normalized creatinine but also because the kidneys have a high capacity to form prostanoids [109,110]. Therefore, it is generally assumed that the stable end metabolites reflect systemic biosynthesis, while more intermediate metabolites rather originate directly from the kidney. Although urinary prostanoid metabolites are synonymously addressed as systemic prostanoids, one should actually distinguish between circulatory and systemic prostanoids, which include kidney-derived prostanoids. For instance, both 6-keto PGF_1α_ and 2,3-dinor-6-keto PGF_1α_ are detectable in human urine, although at different levels. There is a statistically significant but only moderate correlation (r^2^ = 0.55) between these two metabolites of PGI_2_ [48], which indicates that the fraction that is fully metabolized to yield 2,3-dinor-6-keto PGF_1α_ largely represents the circulatory PGI_2_ levels, while the fraction that is only partly metabolized to 6-keto PGF_1α_ likely is a mix of systemic prostanoids that originates from both the circulation and the kidneys. This concept has been recently challenged by a case report [111] of one patient with inherited human group IV A cPLA_2α_ deficiency due to a rare homozygous deletion in the PLA2G4A gene, causing a premature stop codon that results in an almost complete loss of systemic formation of eicosanoids. After suffering end-stage renal failure due to tubulointerstitial nephritis, this patient received an unrelated donor kidney from their spouse, which created a unique human whole-body cPLA_2α_ knock-out, kidney-specific knock-in model. Interestingly, the urinary levels of 2,3-dinor-6-keto PGF_1α_ and 11-dehydro TXB_2_ were restored from very low levels before transplantation to similar and even higher levels, respectively, than those in healthy volunteers [111]. The authors of the case report concluded that the urinary metabolites had purely renal origin [111] and that they poorly reflect production within the circulation in general [112]. Although this conclusion was criticized [113], among other reasons for possible adaptation processes within the kidney after transplantation and for comparing this patient to healthy volunteers instead of other renal failure patients after kidney transplantation, it raises awareness to critically consider established concepts, e.g., to what extent kidney-derived prostanoids are metabolized. Unfortunately, the case report provided only limited data; additional information on urinary levels of 6-keto PGF_1α_ or TXB_2_ and 2,3-dinor TXB_2_ might have been revealing. This report raises furthermore awareness to regard eicosanoid lipid mediators as an interconnected network and evaluate it with appropriate controls on different levels. Ideally, observed changes in one or a few metabolites should be backed-up in a broader lipidomic approach and can be reflected in the expression and activity levels of the involved enzymatic machinery.

## 5. Prostanoid Metabolites as Relevant Biomarkers in Human Disease

The role of primary prostanoids as lipid mediators in disease is well described, e.g., in inflammatory pain [114], autoimmunity [115] and cancer [116], and their potential as a therapeutic target has recently been reviewed [117]. Here, we summarize what is known about the use of prostanoid metabolites as potential biomarkers in human disease. Biomarkers are defined as endogenous molecules that reflect the health status of a person. They are present in biological fluids or soft tissues and change from their baseline concentrations in response to physiological or pathological processes, respectively. Hence, they can be measured as an index of disease. Furthermore, they also respond to pharmacological intervention. Thus, biomarkers can be used as diagnostic, predictive, or prognostic tools to evaluate a patient’s condition or response to treatment.

Prostanoids emerge locally in a cell- and tissue-specific way, their concentration may differ substantially between different tissues, but also change immensely with physiological responses. Therefore, prostanoids can be viewed as biomarkers; however, interpretation of measured concentrations is not trivial.

A few studies investigated primary prostanoids in local tissues for their potential to serve as biomarkers of disease. It should be noted that this approach has some caveats that limit a wide application. Invasive procedures are usually necessary to obtain tissue samples from human subjects. Furthermore, the measured prostanoid levels may differ from the true levels in vivo due to ongoing metabolism. Therefore, studying the biosynthetic capacity of a certain tissue and its changes during disease by investigating induced prostanoid formation after in vitro stimulation might be a better alternative to directly measuring primary prostanoids as biomarkers. In patients with familial adenomatous polyposis (FAP), exogenously added arachidonic acid to mucosa samples induced the formation of all five primary prostanoids. Chemoprevention for colorectal neoplasia with NSAIDs leads to a reduction in the levels of all prostanoids, which correlated with the number and size of adenomas in these patients and could thus be used to monitor the clinical progression of polyposis [118]. In CSF from patients with amyotrophic lateral sclerosis (ALS) [119], infants with hypoxic damage [120], and patients with multiple sclerosis (MS) [121], the levels of PGE_2_ were found to be increased and could serve as a biomarker for disease severity. However, sampling of CSF is invasive, which restricts the use of this biomarker to a relatively small group of patients.

Because sampling of plasma and urine, respectively, is less invasive, these types of samples are more broadly applicable to large patient cohorts and therefore generally preferred to analyze prostanoids as biomarkers for human disease. Prostanoid metabolites detected in plasma/urine and associated with human diseases are summarized in Table 2. Studies on systemic production of prostanoids in humans are mainly performed in urine.

### 5.1. PGE_2_ Metabolites as Biomarkers

As a mainly pro-inflammatory mediator, biosynthesis of PGE_2_ is upregulated by inflammatory stimuli [141,142], which makes PGE_2_ and its metabolites a relevant marker for human diseases that involve chronic inflammation. Plasma levels of PGE_2_ were, however, also investigated as biomarkers in the context of other pathologic conditions, e.g., autism [143], acetylsalicylic acid (ASA)/Aspirin exposure [144], and obesity [31]. PGE_2_ was furthermore measured in saliva and associated with arterial stiffness [126] and markedly correlated with age [145]. Several of these studies used immunoassays that report both PGE_2_ and several of its metabolites combined as a measure of inflammation. Therefore, it is not always clear which specific analyte was actually reported. However, many of these studies claim that it is not important to distinguish between different metabolites. More defined biomarker studies focused on bicyclo-PGE_2_ in plasma and reported it as a marker for malaria pathogenesis [127]. Prostanoids also play a prominent role during pregnancy and labor, which has been recently reviewed [6]. In this context, plasma levels of PGE_2_, 13,14-dihydro-15-keto-PGE_2_, and bicyclo-PGE_2_ yielded, however, inconsistent results [6].

While the measurement of unmetabolized PGE_2_ is uncommon in urine, it was suggested as a biomarker for recurrent urinary tract infections [146]. Urinary metabolites of PGE_2_ on the other hand, foremost tetranor-PGEM, were suggested as potential biomarkers for inflammation in many human diseases, e.g., for ulcerative colitis [122], viral infections in infants [8], and as a prognostic marker for the development of breast cancer [123]. A pre-validation study that investigated tetranor-PGEM as a potential biomarker for the detection of advanced colorectal neoplasia found increased levels of this metabolite in patients with colorectal cancers and large adenomas as compared to healthy patients with small or without adenomas [124]. Importantly, this study also described a sex difference in the levels of tetranor-PGEM with higher levels in men. A subsequent clinical trial with the aim to establish tetranor-PGEM as a biomarker for Crohn’s disease activity (ClinicalTrials.gov Identifier: NCT00496548) was completed in 2015, but results are still pending. Tetranor-PGEM was furthermore shown to be moderately increased in obese subjects and predominantly associated with abdominal obesity and parameters of pre-diabetes [31]. In this study, the combination of tetranor-PGEM with tetranor-PGDM was particularly informative. Both tetranor-PGEM with tetranor-PGDM were also increased in patients with cystic fibrosis [125].

### 5.2. PGD_2_ Metabolites as Biomarkers

Several plasma metabolites of PGD_2_ were investigated in the context of human disease. 13,14-dihydro-15-keto-prostaglandin D_2_ was suggested as a potential biomarker for the diagnosis of nonalcoholic fatty liver (NASH) [133] and serum levels of 11β-PGF_2__α_ as a marker of anaphylaxis [134]. There are a few studies with plasma levels of 15-deoxy-Δ^12,14^-PGJ_2_ in their focus, but only one of them suggested it as a biomarker [34]. In this study, 15-deoxy-Δ^12,14^-PGJ_2_ was increased in the plasma of diabetic patients and inversely correlated to CRP [34].

Biomarker studies on urinary PGD_2_ metabolites are more numerous. Prostanoids, especially PGD_2_, are important mediators released by activated mast cells and thus involved in severe allergic and anaphylactic reactions. Tetranor-PGDM was shown to be increased in patients with food allergies [128] as well as in the urine of patients with Aspirin-intolerant asthma and may be considered a marker for mast cell activation [129]. Recently, the U-BIOPRED (Unbiased Biomarkers for the Prediction of Respiratory Diseases Outcomes) study revealed that the levels of urinary PGD_2_ metabolites (both tetranor-PGDM and 2,3-dinor-11β-PGF_2__α_) were, along with urinary levels of leukotriene (LT)E_4_, increased in patients with mild and severe asthma and associated with a lower lung function. Hence, these urinary eicosanoids were suggested as new, non-invasive biomarkers to detect type 2 inflammation and to phenotype asthma patients on a molecular level [130]. NSAIDs inhibit prostanoid biosynthesis and are a leading cause of drug hypersensitivity reactions [147]. Urinary tetranor-PGDM was suggested as a biomarker for NSAID-induced hypersensitivity [132]. In obese subjects, tetranor-PGDM was markedly increased and associated with serum triglycerides, which led to the hypothesis that elevated levels of systemic PGD_2_ might aggravate obesity [31]. It was also suggested as a urinary marker for the progression of Duchenne muscular dystrophy [131]. Furthermore, urinary tetranor-PGDM was investigated as a marker for atopic dermatitis but was not found to be altered compared to healthy controls [148]. In the field of cancer, tetranor-PGDM was identified as a biomarker in animal studies, e.g., tetranor-PGDM is increased in mice with colitis-associated colorectal cancer [149], but there are no studies in humans.

### 5.3. PGF_2α_ Metabolites as Biomarkers

The in vivo synthesis of PGF_2α_ may be measured in human plasma as 13,14-dihydro-15-keto PGF_2α_ (so-called “PGFM”) [150]. The levels of both PGF_2α_ and PGFM were found to be increased during labor in amniotic fluid, while the same increase was apparent in the blood only for PGFM [6].

Urinary tetranor-PGFM was investigated in humans to study the effect of diet on lipid peroxidation; however, no changes in the levels of this metabolite could be found [151]. In another study, patients with psoriasis were investigated, and tetranor-PGFM was found to be associated with increased psoriasis area and severity index [135].

### 5.4. TXA_2_ Metabolites as Biomarkers

Increased serum levels of TXB_2_ might predict myocardial infarction [152], but metabolites of TXB_2_ in plasma have not been shown to be potential biomarkers for this condition.

While it was previously proposed that 2,3-dinor-TXB_2_ is the main urinary metabolite in mice and 11-dhydro-TXB_2_ is the main urinary metabolite in humans [153], it is now suggested that both metabolites are formed in similar amounts in humans [40]. Their alterations in human disease were recently reviewed [40].

In urine, 2,3-dinor-TXB_2_ is considered a marker of in vivo platelet activation [40]. Urinary 11-dehydro-TXB_2_ was shown to be increased in patients with different stages of chronic kidney disease [136]. This metabolite was also associated with an increased risk of cardiovascular events as measured at baseline in patients with nonvalvular atrial fibrillation [154], and vascular inflammation [138]. Other studies on this biomarker in cardiovascular disease were reviewed [137].

### 5.5. PGI_2_ Metabolites as Biomarkers

Plasma levels of 6-keto PGF_1α_ were investigated during labor; the results were, however, inconsistent [6].

While 6-keto PGF_1α_ can also be detected in urine, the main urinary metabolite that reflects the systemic biosynthesis of PGI_2_ is 2,3-dinor-6-keto PGF_1α_, which is also called PGIM. The ratio between 2,3-dinor-6-keto PGF_1α_ and 11-dehydro-TXB_2_ was suggested to reflect the pathological state in diabetic patients [139] and to evaluate pregnancy after in vitro fertilization [155]. 2,3-dinor-6-keto PGF_1α_ was also suggested as a prognostic marker for acute kidney injury [140].

## 6. Conclusions

The five primary prostanoids, PGE_2_, PGD_2_, PGF_2__α_, PGI_2_, and TXA_2_, are important mediators and play a central role in many physiological and pathophysiological processes. Inherent to their role as evanescent autacoids in vivo, they have a limited action range and rapidly lose their biological activity through metabolic rendering. Pioneering work has elucidated a plethora of metabolites, of which major plasma and urinary metabolites were identified for each of the primary prostanoids. Using sensitive methodology, these metabolites can be measured as substitutes to reflect integrated prostanoid biosynthesis in vivo and to serve as diagnostic, predictive, or prognostic biomarkers for human diseases. However, only a limited number of biomarker studies are available to date that explore the potential of measuring prostanoid metabolites and their fluctuations during pathological changes in humans in relation to disease outcomes.

Metabolism of prostanoids is continuously ongoing; hence, measuring the more stable end products in the urine, i.e., the tetranor-metabolites of PGE_2_, PGD_2_, and PGF_2__α_, and the dinor-metabolites of PGI_2_, and TXA_2_, yields the most consistent and reliable results. However, depending on the specific research question, analysis of other biofluids with their respective major prostanoid metabolites can be informative, too. Urine and plasma are the most commonly studied biofluids for prostanoid metabolite analysis, and they lend themselves to biomarker studies in large cohorts because they are relatively easily sampled. Analyzing prostanoid metabolites in these samples is, however, an indirect approach to estimating systemic in vivo generation of primary prostanoids. Their levels are integrated pools of production throughout the organism and do not reveal any information about specific levels in their tissue(s) of origin. Initial proof-of-principal studies that elucidate the pathogenic mechanism of primary prostanoids under a given disease condition, identify their biological source, trace their metabolism, and verify their induced concentration range as compared to the healthy state, are therefore a prerequisite for the identification of reliable biomarkers and form, combined with larger clinical studies that demonstrate the correlation of systemic levels of their metabolites with specific symptoms, a powerful translational approach.

## Figures and Tables

**Figure 1 metabolites-12-00721-f001:**
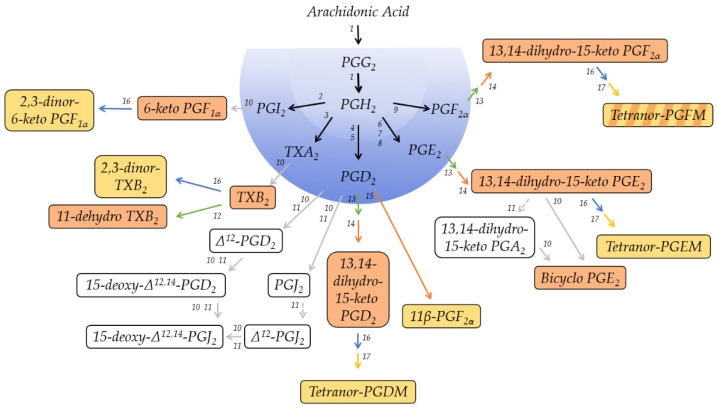
Schematic overview of the prostanoid pathway with the major metabolites that can be detected in human plasma or urine, respectively. The individual metabolites are described in detail in the text. Metabolites on a red background are regarded as major plasma metabolites; metabolites on a yellow background are major urinary metabolites. Red arrows indicate reduction, green, blue, or yellow arrows oxidation, and grey arrows non-enzymatic reactions. 1, Cyclooxygenase-1/2 (COX, EC 1.14.99.1); 2, Prostacyclin Synthase (PTGIS, EC 5.3.99.4); 3, Thromboxane A Synthase (TBXAS1, EC 5.3.99.5); 4, Hematopoietic Prostaglandin D Synthase (HPGDS, EC 2.5.1.18); 5, Lipocalin-Type Prostaglandin D Synthase (L-PGDS, EC 5.3.99.2); 6, Microsomal Prostaglandin E Synthase-1 (MPGES1, EC 5.3.99.3); 7, Microsomal Prostaglandin E Synthase-2 (MPGES2, EC 5.3.99.3); 8, Cytosolic Prostaglandin E Synthase (CPGES, EC 5.3.99.3); 9, Prostaglandin F Synthase (PGFS; EC 1.1.1.188); 10, non-enzymatic degradation; 11, albumin-mediated degradation; 12, 11-hydroxy Thromboxane Dehydrogenase (EC 1.2.1.3); 13, 15-hydroxyprostaglandin Dehydrogenase (15-PGDH, EC 1.1.1.141); 14, 15-oxo-prostaglandin Δ13-reductase (EC 1.3.1.48); 15, NADPH-dependent PGD_2_ 11-ketoreductase (EC 1.1.1.188); 16, enzymatic β-oxidation; 17, enzymatic ω-oxidation.

**Figure 2 metabolites-12-00721-f002:**
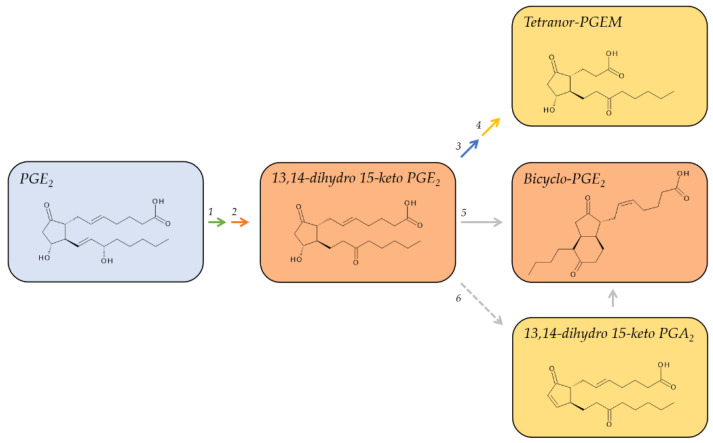
Major metabolic pathway of PGE_2_. Metabolites on a red background are regarded as major plasma metabolites; metabolites on a yellow background are major urinary metabolites. The red arrow indicates reduction, the green, blue, or yellow arrows oxidation, and the grey arrows non-enzymatic/albumin mediated reactions. 1, 15-hydroxyprostaglandin Dehydrogenase (15-PGDH, EC 1.1.1.141); 2, 15-oxo-prostaglandin Δ13-reductase (EC 1.3.1.48); 3, enzymatic β-oxidation; 4, enzymatic ω-oxidation; 5, non-enzymatic degradation; 6, albumin-mediated degradation.

**Figure 3 metabolites-12-00721-f003:**
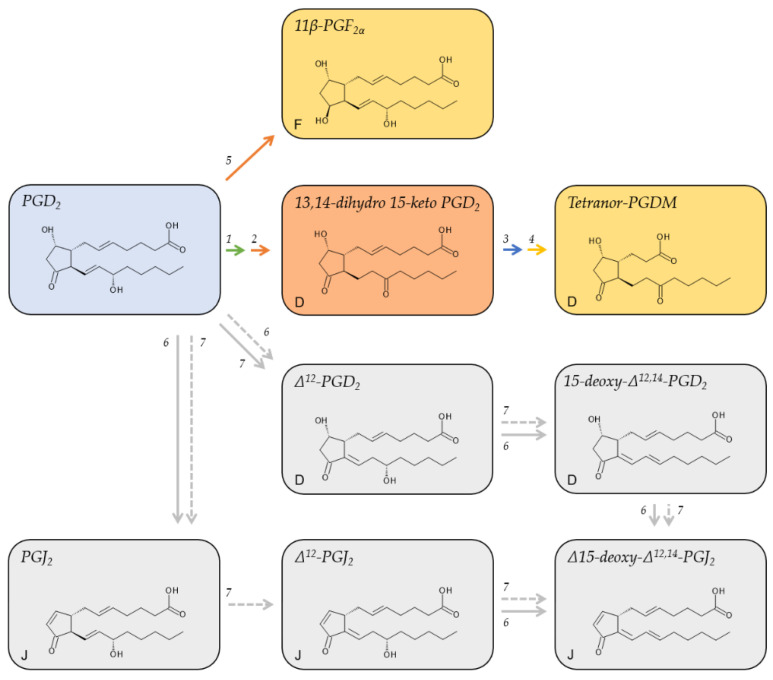
Major metabolic pathways of PGD_2_. Metabolites on a red background are regarded as major plasma metabolites; metabolites on a yellow background are major urinary metabolites; metabolites on a grey background are cyclopentenone derivatives of PGD_2_. Red arrows indicate reduction, green, blue, or yellow arrows oxidation, and grey arrows non-enzymatic/albumin mediated reactions. The letters in the lower left corner denote the ring structure of the respective metabolite. 1, 15-hydroxyprostaglandin Dehydrogenase (15-PGDH, EC 1.1.1.141); 2, 15-oxo-prostaglandin Δ13-reductase (EC 1.3.1.48); 3, enzymatic β-oxidation; 4, enzymatic ω-oxidation; 5, NADPH-dependent PGD_2_ 11-ketoreductase (EC 1.1.1.188); 6, albumin-mediated degradation; 7, non-enzymatic degradation.

**Figure 4 metabolites-12-00721-f004:**
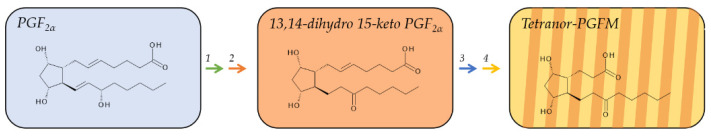
Major metabolic pathway of PGF_2__α_. Metabolites on a red background are regarded as major plasma metabolites; metabolites on a yellow background are major urinary metabolites. The red arrow indicates reduction, and the green, blue, or yellow arrows oxidation reactions. 1, 15-hydroxyprostaglandin Dehydrogenase (15-PGDH, EC 1.1.1.141); 2, 15-oxo-prostaglandin Δ13-reductase (EC 1.3.1.48); 3, enzymatic β-oxidation; 4, enzymatic ω-oxidation.

**Figure 5 metabolites-12-00721-f005:**
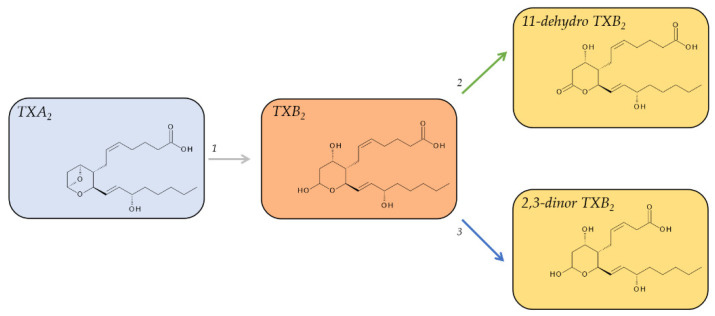
Major metabolic pathway of TXA_2_. Metabolites on a red background are regarded as major plasma metabolites; metabolites on a yellow background are major urinary metabolites. The grey arrow indicates non-enzymatic decomposition, the green and blue arrows oxidation reactions. 1, non-enzymatic degradation; 2, 11-hydroxy Thromboxane Dehydrogenase (EC 1.2.1.3); 3, enzymatic β-oxidation.

**Figure 6 metabolites-12-00721-f006:**
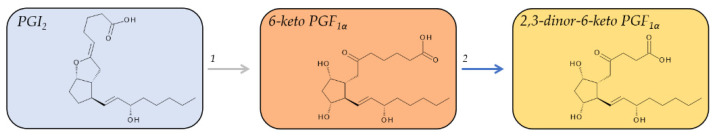
Major metabolic pathway of PGI_2_. Metabolites on a red background are regarded as major plasma metabolites; metabolites on a yellow background are major urinary metabolites The grey arrow indicates non-enzymatic decomposition and the blue arrow oxidation reactions. 1, non-enzymatic degradation; 2, enzymatic β-oxidation.

**Table 1 metabolites-12-00721-t001:** Prostanoid metabolites and their concentrations in biofluids.

Primary Prostanoid	Metabolite	Concentration ^1^	Comment
PGE_2_		Plasma: 3–12 pg/mL	t_1/2_ < 1min in circulation [27].
	13,14-dihydro-15-keto PGE_2_	Plasma: 10–100 pg/mL [28]	t_1/2_ = 9 min (in vivo in dogs) [29]. t_1/2_ = 7 h in protein-free buffer [30]. t_1/2_ = 3 h in diluted plasma [30].
	tetranor-PGEM	Urine: 7–40 µg per 24 h urine collection (µg/day) [31]	Major urinary metabolite of PGE_2_ in humans. Marker for systemic PGE_2_ biosynthesis in vivo. Considered a stable metabolite.
	bicyclo PGE_2_	Plasma: 20–25 pg/mL [28,32]	Non-enzymatically formed, but base-catalyzed. Considered a stable metabolite.
	13,14-dihydro-15-keto PGA_2_	n.d.	Non-enzymatically formed. Considered a stable metabolite.
PGD_2_			t_1/2_ = 0.9 min [33].
	tetranor-PGDM	Urine: 1.5 ± 0.3 ng/mg creatinine [31]	Major urinary metabolite of PGD_2_ in humans. Marker for systemic PGD_2_ biosynthesis in vivo.
	11β-PGF_2__α_	n.d.	
	15d-PGD_2_	n.d.	Bioactive metabolite of PGD_2_; activates PPARγ.
	15-deoxy-Δ^12,14^-PGJ_2_	Plasma: 2 to 350 pg/mL [34] Urine: 6.3 ± 2.7 pg/mg creatinine [35]	
	13,14-dihydro-15-keto PGD_2_	n.d.	Not regularly measured as PGD_2_ metabolite in plasma or urine.
	Δ^12^-PGJ_2_	n.d.	Not regularly measured as PGD_2_ metabolite in plasma or urine.
	PGJ_2_	n.d.	Not regularly measured as PGD_2_ metabolite in plasma or urine.
PGF_2α_			
	13,14-dihydro-15-keto PGF_2__α_	Plasma: 0.08–20 pmol/mL (basal levels) [36]; 40–60 pg/mL (late pregnancy); 1200–4100 pg/mL (parturition) [37]	
	tetranor-PGFM	Plasma: 60–100 pg/mL (late pregnancy); 1000 and 2000 pg/mL (parturition); 100–300 pg/mL (24 h after parturition) [37] Urine: 11–59 µg/day (♂), 7–13 µg/day (♀), 2- to 5-fold increase during pregnancy [38]	Major urinary metabolite of PGF_2α_ in humans.
TXA_2_			t_1/2_ = 30 s in circulation [39].
	TXB_2_	1–2 pg/mL (3–6 fmol/mL) in uninduced blood 300–400 ng/mL (0.8–1.0 nmol/mL) in fully coagulated blood [40]	t_1/2_ = 7 min [41]. Non-enzymatically formed. Detectable in plasma. Stable metabolite ex vivo. Urinary TXB_2_ may reflect intrarenal TXA_2_ production. Serum TXB_2_ reflects capacity of platelets to synthesize TXA_2_ in vitro.
	11-dehydro TXB_2_	Plasma: 0.9–1.8 pg/mL [42] Urine: 0.9–4.3 pg/mL, 30–70 ng/mmol creatinine, 80–190 pmol/mmol creatinine [43]	Detectable in plasma. t_1/2_ = 45 min in circulation [42]. Major urinary metabolite of TXA_2_ in mice.
	2,3-dinor TXB_2_	Urine: 10.3 ng/h (138 pg/mg creatine) 45 pmol/mmol creatinine [43]	Urinary 2,3-dinor TXB_2_ is a marker for systemic TXA_2_ synthesis in vivo.
PGI_2_			t_1/2_ = 30 s (in vivo). t_1/2_ = 1.29–1.52 min (in vivo in cats) [44], t_1/2_ = 6–10 min (in vitro) [45,46,47].
	6-keto PGF_1α_	n.d.	Non-enzymatically formed. Major plasma metabolite of PGI_2_. Urinary 6-keto PGF_1α_ may reflect intrarenal PGI_2_ production.
	2,3-dinor-6-keto PGF_1α_	Urine: 100 pg/mg creatinine [48,49]	Major urinary metabolite of PGI_2_ in humans. Urinary 2,3-dinor-6-keto PGF_1α_ is a marker for systemic PGI_2_ synthesis in vivo.

^1^ Concentrations are reported as found in the literature; therefore, units and normalization approaches may differ. When available, data on prostanoid metabolites in healthy human subjects are reported; however, some information comes from other species. n.d., no data.

**Table 2 metabolites-12-00721-t002:** Prostanoid metabolites as urinary/plasma biomarkers in human disease.

Prostanoid Metabolite	Biological Fluid	Association to Disease
PGE_2_		
tetranor-PGEM	Urine	Ulcerative colitis [122]; Viral infections in infants [8]; Breast cancer [123]; Colorectal cancers [124]; Abdominal obesity/pre-diabetes [31]; Cystic fibrosis [125]
combined PGE_2_ and PGE_2_ metabolites	Plasma/serum/saliva	Obesity [31]; Arterial stiffness [126]; Malaria pathogenesis [127]
PGD_2_		
tetranor-PGDM	Urine	Cystic fibrosis [125]; Food allergy [128]; Aspirin-intolerant asthma [129]; Astma [130]; Obesity [31]; Duchenne muscular dystrophy [131]; Aspirin-exacerbated respiratory disease [132]
2,3-dinor-11β-PGF_2__α_	Urine	Asthma [130]
13,14-dihydro-15-keto-prostaglandin D_2_	Serum	Nonalcoholic fatty liver (NASH) [133]
11β-PGF_2__α_	Serum	Anaphylaxis [134]
15-deoxy-Δ^12,14^-PGJ_2_	Plasma	Diabetes [34]
PGF_2α_		
tetranor-PGFM	Urine	Psoriasis [135]
TXA_2_		
2,3-dinor-TXB_2_	Urine	Acute coronary syndromes [40]
11-dhydro-TXB_2_	Urine	Cardiovascular risk/Acute coronary syndromes [40]; Chronic kidney disease [136]; Cardiovascular disease [137]; Vascular inflammation [138]; Diabetes [139]
PGI_2_		
2,3-dinor-6-keto PGF_1α_	Urine	Diabetes [139]; Acute kidney injury [140]

## Data Availability

The data presented in this study are available in the main article and the Appendix A.

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
