# Peer review of "Prostanoid Metabolites as Biomarkers in Human Disease"

_metabolites, 2022, doi:10.3390/metabo12080721_

Round 1
Reviewer 1 Report
This is a well written solid review on prostanoids as disease biomarkers. The only suggestion is to add discussion on another possible pitfall for prostanoid analysis related to a fast non-enzymatic oxidation of biological material during storage with the production of prostanoid-like molecules (iso-prostains) that cross-react with prostanoinds during immunoassay and LC-MS/MS analysis (e.g. Brose et al, 2011).
Author Response
Thank you for your review and your very important comment regarding isoprostanes. We have added “This includes separation of enzymatically and non-enzymatically formed species. Biological samples may undergo oxidation during prolonged storage, which may create prostanoid-like, non-enzymatically formed isoprostanes [106]” on Line 435; and ”Furthermore, biological samples may undergo oxidation during storage and sample handling, leading to isoprostanes, which are non-enzymatically formed, prostanoid-like molecules that can interfere with enzyme immunoassays and LC-MS/MS analysis, and result in measuring artifacts [106]. “ on line 485; including the suggested reference by Brose et al. ([106]).
We have included a graphical abstract.
Table 1 was inserted close to the section where it first was mentioned instead of in the end of the document.
Track changes have been used.
Reviewer 2 Report
This is a comprehensive review focusing Prostanoid Metabolites as Biomarkers in Human Disease. Importantly, authors emphasize the relevance of the diverse prostanoids, the possibilities to measure them mainly in plasma or urine and their relevance as biomarkers in different diseases. In general, the work summarizes very well the field, however I would just suggest an improvement:
The use of NSAIDs is mention that should be carefully controlled in any study that aims to quantify prostanoid metabolites. In this sense I would suggest to the authors, include in your review few lines illustrating NSAIDs as one of the leading causes of hypersensitivity reactions to drugs, and in turn highlight the relevance of prostanoids in severe allergic reactions and anaphylaxis.
Minor
-A spelling mistake word was found in line 169: phosphorylation. Please change.
-ASA exposure is expressed as abbreviation in line 469. Talking about citation number 116, I can see that you mean Potential anti-inflammatory and anticarcinogenic effects of aspirin (ASA). Please attention because ASA also means active systemic anaphylaxis. Please modify
Author Response
Thank you for your nice comments. We have clarified the abbreviation of ASA on line 661 and corrected the misspelling of phosphorylation on line 201. We have also included a sentence on NSAIDs and hypersensitivity reactions as suggested: However, since we in this review not discuss the importance of primary prostanoids, we have mentioned it in the context of prostanoid (PGD2) metabolites as biomarkers of e.g. anaphylaxis. This was done on Line 696 and Line 707: “Non-steroidal anti-inflammatory drugs (NSAIDs) inhibit prostanoid biosynthe-sis and are a leading cause of drug hypersensitivity reactions [155]. Urinary tetranor-PGDM has been suggested as a biomarker for NSAID-induced hypersensitivity [139].”
We have included a graphical abstract.
Table 1 was inserted close to the section where it first was mentioned instead of in the end of the document.
Track changes have been used.
Reviewer 3 Report
The manuscript fits the Metabolites and will certainly attract attention of even broader audience, especially because it deals with very important topics in a very modern way. Therefore, it deserves considering for publication, but after minor revision, which should include:
Ø tetranor-PGEM Urine: 7-40 μg/day What does it mean? and there is no literature reference
Ø other compounds should also be referenced in Table 1
Ø a contents should be made in order to better understand what the article is about
Ø in chapter Prostanoid Metabolites as Relevant Biomarkers in Human Disease authors should include a table
Ø change the order of chapters: chapter 4 Pitfalls during prostanoid analysis and chapter 5 Prostanoid Metabolites as Relevant Biomarkers in Human Disease
Ø in the figures 2,3,4,5,6 the information on enzymes should also be included in the same way as in figure 1
Ø table 1, and supplementary table 1 it's the same table
Author Response
Thank you for your great review.
In Table 1 we have changed “7-40µg/day” to “7-40 µg per 24h urine collection” and included a reference, and additional references to other compounds have also been added.
A table of contents have been included to give a better overview about what the article is about as suggested by the reviewer.
We have included Table 2 in the section of Prostanoid Metabolites as Relevant Biomarkers in Human Disease and refer to the table on Line 648. Thank you for that great suggestion.
As suggested by reviewer 3 we have changed the order of chapter 4 and 5.
Figures and figure legends have been updated to include information on enzymes.
We submitted a supplementary table that was different to table 1. However, we will upload it again and hope that this with solve the problem.
We have included a graphical abstract.
Table 1 was inserted close to the section where it first was mentioned instead of in the end of the document.
Track changes have been used.